# Integration of In Silico, In Vitro and In Situ Tools for the Preformulation and Characterization of a Novel Cardio-Neuroprotective Compound during the Early Stages of Drug Development

**DOI:** 10.3390/pharmaceutics14010182

**Published:** 2022-01-13

**Authors:** Claudia Miranda, Alejandro Ruiz-Picazo, Paula Pomares, Isabel Gonzalez-Alvarez, Marival Bermejo, Marta Gonzalez-Alvarez, Alex Avdeef, Miguel-Ángel Cabrera-Pérez

**Affiliations:** 1Unit of Modeling & Experimental Biopharmaceutics, Central “Marta Abreu” de Las Villas, Centro de Bioactivos Químicos Universidad, Santa Clara 50100, Cuba; cmiranda@uclv.cu (C.M.); macabrera@uclv.edu.cu (M.-Á.C.-P.); 2Department Engineering of Pharmacokinetics and Pharmaceutical Technology Area, Miguel Hernandez University, 03550 Alicante, Spain; alejandroruizpicazo@gmail.com (A.R.-P.); paula.pomares@alu.umh.es (P.P.); isabel.gonzalez@goumh.umh.es (I.G.-A.); mbermejo@goumh.umh.es (M.B.); 3In-ADME Research, 1732 First Avenue # 102, New York, NY 10128, USA; alex@in-adme.com

**Keywords:** intestinal permeability, Caco-2, solubility-pH, p*K*a, BCS, JM-20, MDCK

## Abstract

The main aim of this work is the biopharmaceutical characterization of a new hybrid benzodiazepine-dihydropyridine derivative, JM-20, derived with potent anti-ischemic and neuroprotective effects. In this study, the p*K*a and the pH-solubility profile were experimentally determined. Additionally, effective intestinal permeability was measured using three in vitro epithelial cell lines (MDCK, MDCK-MDR1 and Caco-2) and an in situ closed-loop intestinal perfusion technique. The results indicate that JM-20 is more soluble at acidic pH (9.18 ± 0.16); however, the Dose number (Do) was greater than 1, suggesting that it is a low-solubility compound. The permeability values obtained with in vitro cell lines as well as with the in situ perfusion method show that JM-20 is a highly permeable compound (Caco-2 value 3.8 × 10^−5^). The presence of an absorption carrier-mediated transport mechanism was also demonstrated, as well as the efflux effect of P-glycoprotein on the permeability values. Finally, JM-20 was provisionally classified as class 2 according to the biopharmaceutical classification system (BCS) due to its high intestinal permeability and low solubility. The potential good oral absorption of this compound could be limited by its solubility.

## 1. Introduction

Drug discovery and development processes are extremely costly in both time and money. Today, the number of new synthesized chemical entities with high biological activity but poor physicochemical and biopharmaceutical properties is increasing, limiting new candidates for prescription drugs [1]. The characterization of these properties in the early stages of drug development is very important to avoid potential failures during clinical trials and to stop further development of the drug product.

The Biopharmaceutics Classification System (BCS) [2] is a widely used scientific framework for waiver of in vivo bioequivalence studies, as well as for decision-making in drug discovery and development [3]. This tool classifies drug substances according to solubility/dosage and permeability values as follows: drugs with high solubility/high permeability (Class 1), drugs with low solubility/high permeability (Class 2), drugs with high solubility/low permeability (Class 3) and drugs with low solubility/low permeability (Class 4). BCS has proven to be a very important tool for the identification of compounds whose solubility and absorption characteristics may be sensitive to physiological and formulation variables, allowing early optimization in the preformulation stages from the knowledge of the physicochemical properties of the active pharmaceutical ingredient (API) [3]. In this sense, it is evident that new compounds with low permeability and solubility/dissolution will have low and highly variable oral bioavailability, which may limit the chances of developing a clinically useful product.

Classification criteria for solubility and permeability properties according to BCS are not fully harmonized among the major regulatory agencies. For instance, to classify a drug’s solubility as high, the World Health Organization (WHO) requires the highest single therapeutic dose to dissolve in ≤250 mL of aqueous medium in the pH range of 1.2–6.8. The United States Food and Drug Administration (US-FDA) and the European Medicinal Agency (EMA) use for solubility classification the highest dosage strength [4,5]. Regarding drug permeability, there is a consensus on the cut-off value for high permeability (oral absorbed fraction fa higher than 85%), as well as on different experimental methods to establish in vivo permeability, such as in vivo or in situ intestinal perfusion studies in animal models and in vitro cells methods (e.g., Caco-2: immortalized cell line of human colorectal adenocarcinoma cells, MDCK: Madin-Darby Canine Kidney Cells). These cell lines are a well-stablished in vitro model for human intestinal permeability, MDCK cells, despite their renal origin, have demonstrated a good correlation with the absorbed human fraction and MDCK-MDR1 and Caco-2 cell lines present the added value of expressing P-glycoprotein [6,7,8,9]. In situ permeability studies in rats, either with the single-pass method or closed-loop perfusion method, are also accepted models for permeability classification. All in vitro and in situ models should be internally validated, i.e., demonstrate good correlation of permeability and fa values for a set of model compounds including low, intermediate and high permeability drugs. Validation must be carried out at least once, and the classification test of any new drug is performed by comparison with the permeability of the high permeability model compound, which in general, is metoprolol. Bearing in mind that each permeability model has advantages, disadvantages and a specific domain of application, a combination of these methodologies will provide a better permeability characterization for any new drug candidate [10].

JM-20 (3-ethoxycarbonyl-2-methyl-4-(2-nitrophenyl)-4,11-dihydro-1H-pyrido [2,3-b][1,5] benzodiazepine) is a new hybrid derivate of benzodiazepine-dihydropyridine with high potentiality for neuroprotection in experimental models related to cerebral ischemia. Several in vitro and in vivo models have been performed [11,12,13] and diverse multitarget mechanisms present in most common neurodegenerative diseases have been demonstrated. Among the most relevant mechanisms are (i) antioxidant and protective activity in brain-derived mitochondria [14], (ii) modulation of the glutamatergic system without amnesic or addictive effects [15], and (iii) anti-inflammatory and anti-apoptotic effects [16]. The combination of more than one neuroprotective pharmacophore in the same chemical structure together with the neuroprotective activity demonstrated in vitro and in vivo suggest that JM-20 may be a good candidate in the treatment of Parkinson’s and Alzheimer’s diseases [17,18,19]. Although this compound has been widely studied from a pharmacological point of view, its high lipophilicity could limit the distribution properties of the compound in the organism after oral administration, affecting the action on the central nervous system.

Given the limited information on the physicochemical and biopharmaceutical properties of JM-20, it is necessary to classify it into BCS to facilitate future drug development. In the present study, an integration of in silico*,* in vitro and in situ methodologies was evaluated for the biopharmaceutical characterization of JM-20 according to CS.

## 2. Materials and Methods

### 2.1. In Silico Studies

#### Physicochemical and Biopharmaceutical Properties

Properties such as molecular weight (MW), polar surface area (PSA), number of rotatable bonds (RTB), hydrogen bond donors (HBD), hydrogen bond acceptors (HBA), calculated n-octanol/water partition coefficient (log P) and the dissociation constant (p*K*a) were predicted for JM-20 using the software MarvinSketch (version 20.3) [20]. Biopharmaceutic properties such as aqueous solubility (log S), Caco-2 permeability (log Papp) and oral bioavailability (F) were predicted using previous published KNIME workflows [21,22].

### 2.2. Experimental Studies

#### 2.2.1. Compound, Reagents and Buffer Solutions

JM-20 was synthesized, purified and characterized as previously reported [23]. It is a reddish, odorless and crystalline powder [23]. This compound was supplied by the Center for Pharmaceutical Research and Development (CIDEM, Havana, Cuba). The rest of the chemicals used during the study were of the highest grade available. All chemicals were purchased from Sigma-Aldrich (Barcelona, Spain).

For the spectroscopic determination of the dissociation constant (p*K*a), the following buffer solutions were used: (a) pH 2: 50 mL of 0.2 M KCl, 13 mL of 0.2 M HCl, and enough distilled water to fill a 200 mL volumetric flask were added; (b) pH 4–5 range: 2.99 g of CH_3_COONa·3H_2_O, 14 mL of 2.0 M CH_3_COOH, and enough distilled water to fill 1 L volumetric flask were added; (c) pH 6–8 range: 50 mL of 0.2 M KH_2_PO_4_, 22.4 mL of 0.2 M NaOH, and enough distilled water to fill a 200 mL volumetric flask were added; (d) pH 9–10 range: 50 mL of 0.2 M H_3_BO_3_, 32.1 mL of 0.2 M NaOH, and enough distilled water to fill a 200 mL volumetric flask were added; and (e) pH 11–12 range: 100 mL of 0.05 M Na_2_HPO_4_, 8.2 mL of 0.1 M NaOH, and enough distilled water were added to fill a 200 mL volumetric flask. The JM-20 stock solution was prepared at 500 μg/mL in DMSO (≤2% v/v) so as not to alter the p*K*a value [24].

The pH measurements were performed using a sensiON™ (PH31) pH meter. Standard solutions of JM-20 (10 g/mL were prepared adding 0.2 mL of the stock solution in 9.8 mL of each buffer. UV-VIS spectra were recorded on a UV-Vis Spectrophotometer (Thermo Fisher GENESYS™ 10S, Shanghai, China) between 200 and 400 nm (2 nm of resolution). All spectrophotometric data were processed in Microsoft Excel (2018).

For the solubility study, six buffer solutions (pH 1.2, 3.0, 4.0, 4.5, 6.8 and 7.4) were prepared following the protocols published in the USP XXXVIII, United States Pharmacopoeia [25], as described above.

#### 2.2.2. Determination of the Dissociation Constant (pKa)

The experimental p*K*a was determined using the spectrophotometric method proposed by Salgado et al. [26]. The absorbance spectra of the JM-20 were acquired at different pH levels. The wavelengths of maximum absorbance were determined at each pH. Using a plot of observed absorbance versus pH at these analytical wavelengths (262, 316 and 355 nm), the p*K*a values were calculated. To determine these points, the linear equations of the two points closest to the crossing at each curve were solved. The whole procedure was carried out in triplicate and at room temperature (23 ± 2 °C).

#### 2.2.3. Thermodynamic Solubility Determination

Solubility experiments for JM-20 were carried out according to the main regulatory guidelines for biowaivers [27]. The tests were performed at the three recommended pH values (pH 1.2, 4.5 and 6.8) as well as pH 3.5, 4.0 and 7.4 (USP buffer formulations). The shake-flask saturation method proposed by Baka et al. [28] was used in the solubility study and the recommendations of Avdeef et al. were followed to improve the quality of the solubility data [29]. The temperature was regulated at 37 °C. Equilibration times were 6 h stirring + 18 h sedimentation. At the end of equilibration, the two phases were separated by careful decantation. The samples were analyzed with a Thermo Fisher UV-Vis Spectrophotometer (λmax = 262 nm). A linear range of 2–20 μg/mL was obtained (r^2^ > 0.99). Once the solubility range was determined, the dose number (Do) was calculated following Equation (1)
(1)$$\mathrm{Do}=\left(\frac{D}{\mathrm{Vo}}\right)/\mathrm{Cs}$$
where D is the highest strength or highest dose administered (mg) according to FDA or EMA criteria, Vo is the volume of water (250 mL) and Cs is the maximum aqueous solubility (mg/mL). Compounds with Do ≤ 1 or Do > 1 are classified as high or low solubility drugs, respectively.

#### 2.2.4. Refinement of Intrinsic and Salt Solubility

To develop a solution speciation model, the experimental data of log S—pH were used as input into the computer program *p*DISOL-X [30]. The computational algorithm considers the contributions of all species present in solution, including all buffer components. The program derives its own implicit solubility equations internally, given any practical number of equilibria and estimated constants. The equilibrium constants are subsequently refined by weighted nonlinear least-squares regression [31]. The method can test for the presence of specific buffer drug species. The program assumes an initial condition of a suspension of the solid drug in a solution, ideally with the suspension saturated over a wide range of pH. The software calculates the species distribution as a consequence of a sequence of additions of standardized strong-acid titrant HCl to simulate the suspension pH-speciation down to pH ~ 0, the staging point for the subsequent operation. A sequence of perturbations with standardized NaOH is simulated and the solubility is calculated at each point (in pH steps of 0.2), until pH ~ 13 is reached. The ionic strength is rigorously calculated at each step, and p*K*a values (as well as the solubility products and the aggregation and complexation constants) are adjusted accordingly [31]. At the end of the pH-speciation simulation, the calculated log S vs. pH curve is compared to log S vs. pH measured. A log S-weighted nonlinear least squares refinement begins to refine the proposed equilibrium model, using analytical expressions for the differential equations. The process is repeated until the differences between calculated and measured log S values reach a minimum [31].

### 2.3. Permeability Determination

#### 2.3.1. Cell Culture and MDCK, MDCK-MDR1 and Caco-2

MDCK (Madin–Darby Canine Kidney) cells, their transfected clone MDCK-MDR1, and Caco-2 cells were grown in the modified Dulbecco Eagle medium added with L-glutamine, fetal bovine serum and penicillin−streptomycin. The number of cells seeded in transwells with a surface area of 4.2 cm^2^ depended on the cell line: 250,000 cells/cm^2^ for MDCK and Caco-2 and 150,000 cells/cm^2^ for MDCK-MDR1. Experiments were performed between 7 and 9 days for MDCK and MDCK-MDR1 and between 19 and 21 for Caco-2 cells after seeding. Prior to drug placement, cell monolayer integrity was assessed by measuring transepithelial electrical resistance (TEER). Transport studies were performed on an orbital shaker at 37 °C and 50 rpm. Four samples of 200 μL were taken from the receptor side at 0, 15, 30, 60 and 90 min, and two samples from the donor compartment at the beginning and end of the experiment. A standard experiment was performed with the same buffer for the apical and basolateral compartments in the apical to basolateral (A-B) and basolateral to apical (B-A) directions. The drug concentration was 100 μM in all experiments and samples were analyzed immediately by HPLC. The retention time was 2.3 min at maximum wavelengths (λmax = 260 nm). The conditions used for HPLC analyses were: mobile phase composed of a mixture of 50% H_2_O (95% H_2_O + 0.5% trifluoroacetic) and another 50% HPLC-grade methanol. The stationary phase composed of a Phenomenex® KJ0-4282 column, with two 2 μm filters filled with C-18 microparticles of 40 μm size and a Nova Pak C-18 stainless-steel column of 150 mm length. The method was validated and demonstrated to be adequate regarding linearity (r^2^> 0.999), accuracy (relative error <5%), precision or repeatability (SD ≤ 2%), stability (recovery = 98–102%), filter influence (recovery = 98–102%) and specificity (interference < 2%). The lower limit of quantitation of JM-20 was adequate to interpolate the values obtained in the samples.

The apparent permeability coefficient was calculated according to the following Equation (2):(2)$$\begin{array}{cc} C_{receiver,t}= & \frac{Q_{total}}{V_{receiver}+V_{donor}}+\left(\left(C_{receiver,t-1}\cdot f\right)-\frac{Q_{total}}{V_{receiver}+V_{donor}}\right)\\ & \cdot e^{-Peff0,1\cdot S\cdot \left(\frac{1}{Vreceiver}+\frac{1}{Vdonor}\right)\cdot \Delta t} \end{array}$$
where *C_receiver_,t* is the concentration of the drug in the receiver compartment at the time t, *Q_total_* is the total amount of drug in receiver and donor chambers, *V_receiver_* and *V_donor_* are the volumes of receiver and donor chamber, respectively, *C_receiver,t−_*_1_ is the concentration of the drug in receiver compartment at the previous time, *f* is the replacement dilution factor of the sample, *S* is the surface area of the monolayer, Δ*t* is the time interval and *P_eff_* is the coefficient of permeability, as was described by Mangas-Sanjuan et al. [32] All calculations were developed in Microsoft^®^ Excel (2018).

#### 2.3.2. In Situ Perfusion Method

The animal experiments, performed on 250–300 g male Wistar rats, were designed following the guidelines described in EC Directive 86/609, Council of the Europe Convention ETS 123 and Spanish national laws about the use of animals in research, and were approved by the Spanish Committee (Spain, code A1330354541263).

The “closed-loop” in situ perfusion method, based on the Doluisio technique, was performed according to published reports [33]. Briefly, in anesthetized rats in which an abdominal incision was made, the small intestine was located and the bile duct was ligated avoiding enterohepatic circulation of the drug. The procedure for making three loops in the duodenum, jejunum, and ileum was the same as previously described [34]. Intestinal contents were removed by washing with physiological solutions and the peritoneal cavity was covered with cotton wool pads to prevent evaporation of peritoneal fluid and loss of body heat [35,36].

A solution of JM-20 (100 µM) was perfused with one syringe, and 5 min later the sample was collected with the other syringe. This procedure was performed alternatively for 30 min to ensure mixing of the solution in the lumen of small intestine [37,38]. Samples were immediately analyzed for JM-20 at maximum wavelengths (λmax = 262 nm) by HPLC (Alliance-Waters 2695) using a Nova-Pak C18 column (4 μM, 3.9 × 150 mm) and UV detector (Waters 2487). The analytical procedure was as follows previously validated [39]. The exact length of the intestine was determined at the end of the study.

The calculation of permeability takes into account that water flux during the experiment can be significant [40], the drug concentration in each sample was modified according to the following Equation (3):(3)$$C=C_{e}\left(\frac{V_{T}}{V_{0}}\right)$$
where *C_t_* represents the concentration without water reabsorption at time t, *C_e_* is the experimental concentration value, *V_0_* is the initial volume and *V_t_* is the final volume. The coefficient of absorption rate (*k_a_*) and the initial concentration of JM-20 available for absorption (*C*_0_) were determined by non-linear regression of the following Equation (4) [41]:(4)$$C_{t}=C_{0}\cdot e^{-k_{a}t}$$

Permeability value was calculated considering the length of the intestinal segment, in other words, the effective intestinal radius (*r*) considering 10 mL of perfusion volume by the following ratio (Equation (5)):(5)$$Peff=r\cdot \frac{k_{ap}}{2}$$

### 2.4. Statistical Analysis

Statistical results are showed as mean ± standard deviation. Mean values of two groups were analyzed using Student’s two tails *t* tests. A probability level of *p* < 0.05 was established as the significance criterion. The software used for the data analysis was SPSS v 22 (IBM, New York, NY, USA) licensed by the Miguel Hernández University.

## 3. Results

### 3.1. In Silico Physicochemical and Biopharmaceutical Properties

All physicochemical and biopharmaceutical parameters predicted by different computational algorithms are presented in Table 1. As can be appreciated, several physicochemical properties are related to Lipinski rule of five [42] and Veber’s rule [43]. Both rules provide information about the potential absorption and oral bioavailability of drugs.

### 3.2. Determination of the Dissociation Constant (pKa)

Table 1 also represents the predicted p*K*a values for the JM-20 ampholyte molecule. Experimental p*K*a values obtained for JM-20 with a UV-spectrophotometric method were p*K*a1 = 5.48 ± 0.19 (base) and pKa2 = 10.49 ± 0.17 (acid) at 23 °C and ionic strength of approximately 0.1 M. Similar results were recently obtained by Martínez et al. with a p*K*a value of 5.25, using potentiometric titration [44]. The measured p*K*a values for JM-20 were comparable to the calculated p*K*a values.

### 3.3. Solubility Determination

Figure 1 shows the results of the experimental log S-pH profile for JM-20 (see Table 2) compared to the curve corresponding to the Henderson-Hasselbalch (HH) equation. The results evidence the typical U-shaped curve for ampholyte drugs. The predicted Henderson-Hasselbalch curve and the log S-pH profile for JM-20 overlap for pH > 4.5. Above pH 10, solubility gradually increases. Below pH 4.5, the profile is due to chloride salt formation. For pH < 2 the line downward, suggesting the potential “common-ion” effect [31]. The intrinsic solubility for JM-20 is 12.0 μg/mL.

The solubility of JM-20 at pH 1.2, 3.5, 4.0, 4.5, 6.8 and 7.4, as well as the dose/solubility ratio for the minimum dose tested in animals (2 mg/ kg) [23], considering 70 kg as the average weight of a healthy adult, are shown in Table 2. As can be seen, the highest solubility was observed at pH 4.5, however the dose number (D_0_) is higher than 1, confirming that JM-20 is a low solubility drug. The log P value determined in silico was 3.46 (see Table 1), indicating that JM-20 is a lipophilic compound.

Martinez et al. [44] developed a qualitative stability study of JM-20 in aqueous medium at pH values of 1.2, 4.5 and 6.8, during 48 h at 37 ± 1°C. At 24 h some degradation appears at pH 4.5 and 6.8, but the retention time of JM-20 is the same, and only a decrease in the corresponding areas is observed, so it can be deduced that it has not yet undergone changes in its chemical structure.

According to the product owner (technical report), JM-20 is stable in DMSO and absolute ethanol.

### 3.4. Permeability Determination

#### 3.4.1. MDCK, MDCK-MDR1 and Caco-2

In this work, JM-20 permeability was evaluated in three of the most frequent and successful preclinical experimental procedures for predicting intestinal permeability in humans, MDCK, MDCK- MDR1 and epithelial Caco-2 cell cultures. In the case of MDCK- MDR1, selection was based on its ability to express P-glycoprotein (Pgp) [9]. The results obtained in each in vitro cell model are shown in Table 3. As can be seen, the permeability assay for JM-20 was developed in apical-basolateral and basolateral-apical direction, to evaluate the role of passive, active or facilitating transport and efflux mechanisms.

Figure 2 shows a clear comparison of the permeability values obtained for JM-20 in each cell line with those obtained for metoprolol under the same experimental conditions.

#### 3.4.2. In Situ Perfusion Method

Effective intestinal permeability (*P_eff_*) data for JM-20, using the Doluisio perfusion model, were evaluated in three segments of the small intestine and the results are presented in Table 4. The effective permeability of JM-20 had the following rank order of duodenum > jejunum ≈ ileum. The mean permeability coefficient of this compound was comparable to the reported value of metoprolol used as a high permeability reference compound (see Table 4). This comparison was used to classify this compound according to the BCS.

### 3.5. Biopharmaceutical Classification of JM-20

Different provisional biopharmaceutical classification systems have been described in the literature [45,46,47,48,49]. They mainly use alternative methodologies to predict intestinal permeability. Following the proposal of Kasim et al., JM-20 is a BCS Class 2, based on the aqueous solubility value at room temperature and in silico partition coefficients (log P and ClogP) [45]. JM-20 is also classified as BCS class 2 compound by the method proposed by Pham-The et al. where a provisional biopharmaceutical classification was developed based on solubility data (dose number, Do) and predicted or measured permeability in Caco-2 cells [48]. At the same time, permeability values of JM-20 obtained with in vitro cell lines and in situ perfusion in rat were considered for the biopharmaceutical classification of this compound. All details of the provisional biopharmaceutics classification are shown in Table 5.

The clinical repercussion is the human fraction absorbed calculated with Permeability values through Caco-2 monolayers and rat intestine. Results of predictions are summarized in Figure 3.

## 4. Discussion

The in silico prediction and experimental assessment of the physicochemical properties of active compounds during the early stages of drug and formulation development is quite relevant for estimating the biopharmaceutics class and in vivo behavior of oral drugs.

As can be appreciated in Table 1, the space of predicted physicochemical properties evidenced that JM-20 fulfilled Lipinski’s rule of five (MW < 500, log P < 5, HBD < 5 and HBA < 10), and the Veber’s rule (PSA < 140 A and RBN < 10), suggesting the good absorption potential of this molecule [42], and a probable good oral bioavailability [43]. The predicted values for p*K*a (see Table 1) are typical for ampholytes, a kind of molecule with probable poor oral absorption [51]. Nevertheless, the predicted biopharmaceutical properties should be analyzed in details. The predicted aqueous solubility value (2.54 µg/mL) classified JM-20 as a practically insoluble compound, according to USP solubility classification criteria [25]. Meanwhile, the predicted Caco-2 permeability value (7.34 × 10^−6^cm/s), classified JM-20 as a poorly [52,53] or moderately permeable compound [54]. According to these results, JM-20 can be classified as a compound belonging to class II/IV of the BCS, suggesting the evaluation of the influence of absorptive and/or efflux transporters on the absorption process. On the other hand, the low solubility will limit the concentration of JM-20 coming into enterocytes, affecting the fraction of orally absorbed compound. In this sense, the low value of human oral bioavailability (F<50) predicted for JM-20 [22], is in relation to the predicted solubility and permeability values.

The experimental p*K*a values of JM-20 agree with in silico predictions (see Table 1). The p*K*a1 value of 5.48 is attributable to protonation of the nitrogen in position 4 of the benzodiazepine ring while the p*K*a2 value of 10.49 is attributable to deprotonation of the sp3-carbon of the dihydropyridine ring.

As indicated in Figure 1, the solubility data below pH 4.5 is consistent with the formation of a hydrochloride salt. However, it was not possible to fit the solubility value at pH 1.2 with this salt form. For pH > 5 the precipitate is the uncharged ampholyte. Since the p*K*a values were determined at 23 °C and the solubility measurements were performed at 37 °C, the measured p*K*a values were transformed to match the temperature of the saturated solutions (i.e., at 37 °C, the p*K*a values were estimated to be 5.54 and 10.33 [55].

Considering the lowest dose administered in experimental animal studies (2 mg/kg), as well as the normal weight of a healthy adult (70 kg), the final dose used for the calculation of the D/S ratio was 140 mg. The calculated Do for JM-20 at different pH values was greater than 1, which classifies this compound as a poorly soluble drug.

This result is consistent with statistics indicating that 60 to 70% of new compounds under development in the pharmaceutical industry have limited aqueous solubility, and in certain drug categories this percentage can be as high as 90% [56]. Similarly, low-soluble compounds account for 40% of the top 200 oral drugs marketed in the U.S., and the water-insoluble or low-soluble categories of the U.S. Pharmacopeia account for more than one-third of all drugs [57]. Consequently, the classification obtained for this new drug-candidate is in line with the current situation on the pipeline in pharmaceutical companies. The absorption of the oral fraction was predicted based on previous data form our laboratory and the conclusion was that a good correlation was obtained between the two. Miyake et al. obtained the correlation between using chamber data and human data [58,59].

For this type of compounds, different formulation strategies should be developed to improve the poor drug solubility, dissolution rate and oral bioavailability [60,61,62].

In this work, the experimental permeability of JM-20 was determined in three in vitro models such as MDCK, MDCK-MDR1 and Caco-2 cell lines and using the in situ “close-loop” perfusion method in rats.

According to the results obtained in all cell lines (See Table 3), JM-20 can be considered a carried mediated absorption compound. In MDCK cells, a higher permeability is obtained for the A-B direction compared to the B-A direction, which means that the passage of the drug across the membrane is favored by an active or transporter-facilitated mechanism. Therefore, in MDCK cells, JM-20 crosses the membrane by passive diffusion and also involves a transporter that aids the passage of the drug from the gut into the blood.

In MDCK-MDR1 cells permeability in the A-B direction is higher than permeability in the B-A direction as in the other cell type (significant differences *p* < 0.05). If the drug only passed by passive diffusion, the permeability A-B and B-A would have to be similar, that is, if the transports coincide it is because there is only transport by passive diffusion but being different implies the existence of transporters. On the other hand, if the B-A permeability of the two cell types is compared, it is observed that in the MDCK-MDR1 type the permeability is higher (significant differences at *p* < 0.05). This is due to the fact that, the JM-20 is a substrate of the P-glycoprotein transporter (MDR1), which is expressed in this cell line and, therefore, it promotes the passage of the drug to the apical compartment. Comparing the values obtained in the A-B direction, the previous conclusion is corroborated, with a slightly greater permeability in MDCK cells, since resistance due to the secretion transporter is not present, but no statistical differences were detected.

However, when comparing the permeability values (A-B) obtained in both MDCK cell lines with the permeability value of metoprolol (See Figure 2), it is possible to verify that although the compound is a substrate of a secretion transporter, the absorption transporter must have greater affinity for the drug, therefore according to the effective permeability obtained, JM-20 can be classified as a highly permeable drug.

In the case of Caco-2 cells, results similar to those of MDCK cells were obtained. In vitro bidirectional transport of JM-20 in Caco-2 cells showed that the absorptive transport of this compound was greater than its secretory transport, with an efflux ratio of 0.13. The calculated absorptive permeability of JM-20 in the Caco-2 cell monolayer was 18.50 × 10^−5^ cm/s, suggesting efficient permeability of this novel compound [63].

The high permeability value obtained in the Caco-2 model is consistent with the permeability value obtained in the rats. As can be seen in Figure 3, which represents the fraction absorbed versus permeability values relationship in both absorption models, the permeability values of JM-20 fit perfectly with the correlation validated previously. Lozoya-agullo et al. [64,65] have previously demonstrated the excellent correlation between the permeabilities of the Caco-2 model and the rat intestinal permeabilities obtained with the closed-loop perfusion technique in rat and the ability of both models to predict oral fraction absorbed in humans. The new compound assayed JM-20 confirms that permeability values greater than 1.0 × 10^−5^ cm/s in Caco-2 and 6.0 × 10^−5^ cm/s in rat guarantee a nearly complete fraction absorbed.

Considering that JM-20 is a neuroprotective drug and that the purposes of the treatments are chronic diseases, it is interesting to know in depth the permeability of this compound in the intestinal tract since for a chronic disease the oral route of administration may facilitate patient compliance. For this purpose, the in situ intestinal perfusion method was applied in rats. This technique has demonstrated its high correlation (r^2^>0.80) with permeability values in humans for substances with a combined absorption process: passive diffusion and carrier-mediated mechanism [66,67], as well as being considered a good model for the prediction of intestinal permeability in humans [10].

The permeability of JM-20 in the duodenum was higher than that of the other two intestinal segments, suggesting that this compound could be well absorbed throughout the gut with the main site of absorption in the duodenum. For the jejunum and ileum segments, permeability values were similar but lower compared with the duodenum, which could be related to higher levels of Pgp expression in these segments that reduce the permeation of JM-20 across the intestinal membrane.

The permeability values obtained in the different intestinal sections (See Table 4) corroborate the results obtained in the cell lines. For all the “High permeability” compounds in human (Fabs > 85%), with differential permeability along the small intestine, showed at least in one segment a higher permeability value than metoprolol in that same segment. This supports that if a compound has a high absorbed dose fraction, it will have a high-permeability, not necessarily in the jejunum, but somewhere along the relevant intestinal absortive regions [64,68]. Correlations between permeability values in the rat small intestine (whole or segments) obtained by Doluisio’s method and their correlation with the orally absorbed fraction in humans were previously published [64,65]. In fact, Fabs can be calculated considering the permeability values in all segments (whole small intestine) and the estimated oral absorbed fraction was 85.12%. According to this property, the drug may be a good candidate to be formulated for oral administration.

Once the experimental results obtained have been analyzed, we can assure that JM-20 has a low solubility and high intestinal permeability, therefore it belongs to Class 2 of the BCS. The absorption of the drug in this case is limited by its solubility, being this the factor to be improved during the pre-formulation of this API for oral administration.

Many drugs belong to BCS Class 2 are weak bases, with higher solubility at low pH values in the stomach, but lower solubility at near neutral pH in distal segments of the small intestine. These drugs can dissolve at gastric pH facilitating ionization, but their absorption is quite low in the stomach. However, in the case of small intestine, where most drugs are absorbed, higher pH values and lower solubility may affect oral absorption [69].

## 5. Conclusions

In this study, a pH-dependent solubility assay of JM-20 was performed. The p*K*a and solubility values confirm that this weakly basic compound has a low pH-dependent solubility; thus, in the physiological pH range, the dose is not expected to dissolve completely, and dissolution is likely to be the rate-limiting process of its absorption. In vitro permeability assays suggest that an absorption transport mechanism exists in the permeation of JM-20 across the membrane and that this compound is also a substrate of P-glycoprotein. The application of the Doluisio closed-loop in situ perfusion technique in rats allowed to evaluate the effective permeability of the compound, demonstrating the high permeability of this drug compared to metoprolol. Finally, this new chemical entity was classified as BCS class 2 due to its low solubility and high intestinal permeability. These results provide the fundamental basis for the development of an oral formulation that will be evaluated in preclinical trials and will allow further development of this molecule as a neuroprotectant against cerebral ischemia.

## Figures and Tables

**Figure 1 pharmaceutics-14-00182-f001:**
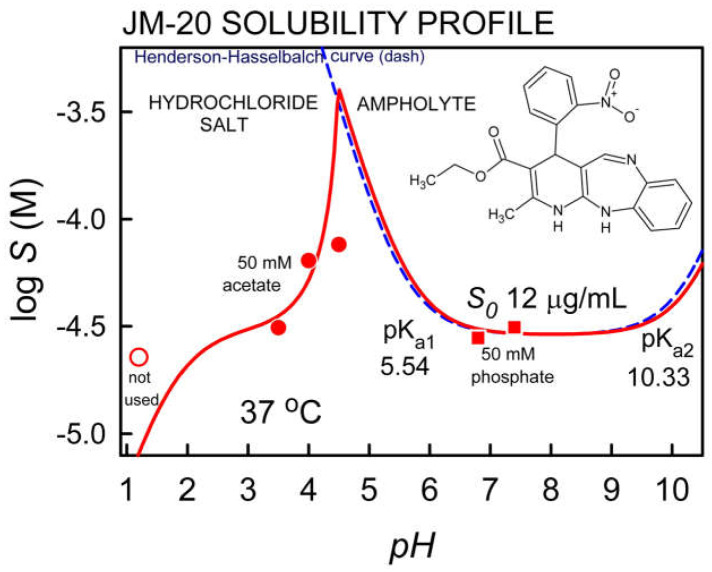
pH-dependent solubility profile of JM-20 (points are solubility values measured by shake-flask method; dashed-line is the theoretical HH curve).

**Figure 2 pharmaceutics-14-00182-f002:**
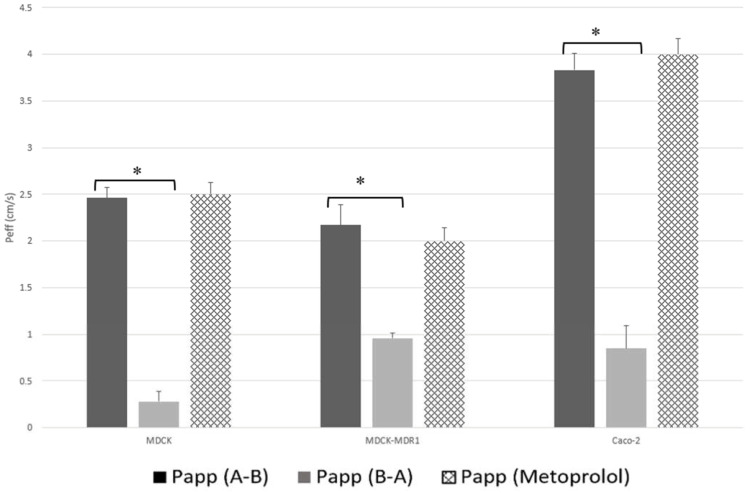
Comparison of apparent permeability values (Papp × 10^−5^ cm/s) measured for JM-20 using MDCK, MDCK-MDR1 and Caco-2 cell lines. Results were compared with permeability value of metoprolol. * Significant differences at *p* < 0.05.

**Figure 3 pharmaceutics-14-00182-f003:**
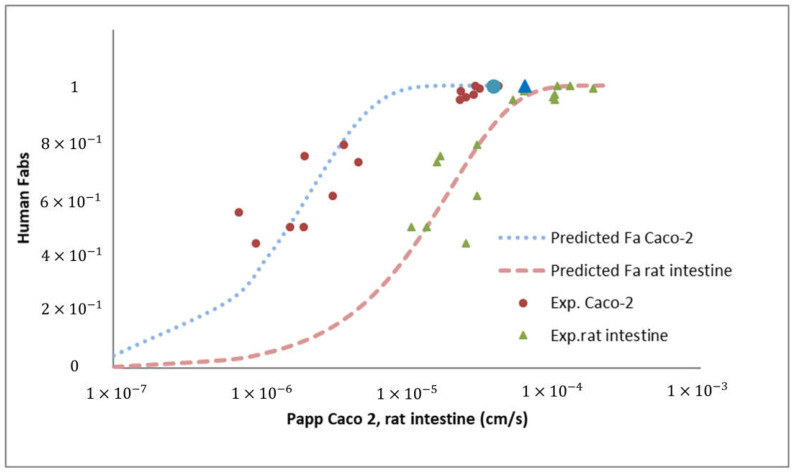
Human fraction absorbed vs. Permeability values in Caco-2 monolayers (apical to basal) or rat intestine. Blue circle and triangle correspond to JM-20 Caco-2 and rat intestine permeability values. Dotted lines correspond to our validated correlation between both techniques and human fraction absorbed [50].

**Table 1 pharmaceutics-14-00182-t001:** Physicochemical and biopharmaceutical predicted properties for JM-20.

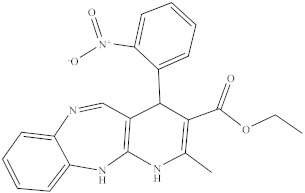	
Property	Predicted Value
Molecular weight (MW)	404.4
n-Octanol/water partition coefficient (log P)	3.46
Polar Surface Area (PSA)	105.86
Number of Rotatable Bonds (RBN)	5
Hydrogen Bond Donors (HBD)	2
Hydrogen Bond Acceptors (HBA)	6
Dissociation Constant (p*K*_a_)	5.05; 11.5
Aqueous solubility (pH 7.4, 37 °C)	2.54 µg/mL
Caco-2 permeability	7.34 × 10^−6^ cm/s
Human oral bioavailability (F%)	<50

JM-20: Structural formula of (3-ethoxycarbonyl-2-methyl-4-(2-nitrophenyl)-4,11-dihydro- 1H-pyrido[2,3-b][1,5] benzodiazepine).

**Table 2 pharmaceutics-14-00182-t002:** Experimental solubility values and number of doses for JM-20.

Buffer Solution	Solubility ^a^ ± SD (μg/mL)	Do ^b^	Solubility ^c^ (μg/mL)	So ^d^ (μg/mL)
1.2	9.18 ± 0.16	61.00	3.3	12.0
3.5	12.72 ± 0.83	44.03	14.3	
4.0	25.38 ± 0.66	22.06	18.9	
4.5	30.84 ± 1.29	18.16	74.9	
6.8	11.30 ± 0.69	49.56	12.5	
7.4	12.64 ± 0.36	44.30	11.9	

^a^ The solubility values were determined at 37 °C, ^b^ The maximum dose was calculated using the minimum dosed tested on animals (2 mg/kg) considering 70 kg as the average weight of a healthy adult. D = 140 mg. ^c^ Solubility calculated by pDISOL-X at 37 °C. ^d^ Intrinsic solubility calculated by pDISOL-X at 37 °C.

**Table 3 pharmaceutics-14-00182-t003:** Apparent permeability coefficients (Papp) for JM-20 using MDCK, MDCK-MDR1 and Caco-2 cell lines (37 °C, stirring at 50 rpm).

Cell Line	P_app_(A-B) (×10^−5^ cm/s)	P_app_(B-A) (×10^−5^ cm/s)
MDCK-MDR1	2.17 ± 0.22	0.96 ± 0.05
MDCK	2.46 ± 0.11	0.28 ± 0.71
Caco-2	3.83 ± 0.18	0.85 ± 0.24

**Table 4 pharmaceutics-14-00182-t004:** Effective permeability coefficients (*P_eff_*) and absorption rate coefficient values for JM-20 using in situ rat intestinal perfusion model.

Compound	*P_eff_* ± SD (×10^−5^ cm/s)		
	Duodenum	Jejunum	Ileum
JM-20	9.20 ± 0.89	6.63 ± 1.30	6.82 ± 1.44
Metoprolol	-	6.23	9.5

Results are shown as mean ± SD (*n* = 4); Permeability values for metoprolol in the same conditions.

**Table 5 pharmaceutics-14-00182-t005:** A provisional BCS classification for JM-20, based on different in situ*,* in vitro and in silico permeability models.

Compound	D_0_ ^a^	BCS_logP_ ^b^ [45]	BCS_ClogP_ ^c^ [45]	BCS_MDCK_	BCS_Caco-2_	BCS_QSPeR_ [48]	BCS_rat_	BCS_Global_
JM-20	61	2	2	2	2	2	2	2

^a^ The dose number was calculated with the lowest aqueous solubility; ^b^ The logP calculated was 3.46 that was greater than logP of Metoprolol (1.72); ^c^ The ClogP calculated was 3.11 that was greater than ClogP of Metoprolol (1.35).

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
