# Peer review of "Integration of In Silico, In Vitro and In Situ Tools for the Preformulation and Characterization of a Novel Cardio-Neuroprotective Compound during the Early Stages of Drug Development"

_pharmaceutics, 2022, doi:10.3390/pharmaceutics14010182_

Round 1

Reviewer 1 Report

You reviced well than before.  I pointed out a question.

Major

You need some comments to compare between the result from Caco-2 and the result from rat perfusion study. That must be important on this journal.

Author Response

You need some comments to compare between the result from Caco-2 and the result from rat perfusion study. That must be important on this journal.

We acknowledge the recommendation and the following paragraph has been included in the discussion

The high permeability value obtained in the Caco-2 model is consistent with the permeability value obtained in rat. As it can be seen in Figure 3, which represents the fraction absorbed versus permeability values relationship in both absorption models, JM20 permeability values fits perfectly to the previously validate correlation. Lozoya-agullo et al. have demonstrated previously the excellent correlation between the Caco-2 model permeabilities and the rat intestinal permeabilities obtained with the closed loop perfusion technique in rat and the ability of both models to predict oral fraction absorbed in humans. The new assayed compound JM20 confirms that permeability values higher than 1.0e-5 cm/s in Caco-2 and 6.0e-5 cm/s in rat guaranties an almost complete fraction absorbed.

Reviewer 2 Report

The manuscript “Integration of in silico, in vitro and in situ tools for the Biopharmaceutical Characterization of a Novel Neuroprotective Compound During the Early Stages of Drug Development” is well designed and interesting in the field of drug discovery for treatment of CNS disorders. The author addresses the some of the below concerns.

In the abstract and introduction, the authors mentioned anti-ischemic and neuroprotection activity. Rewrite the title with the inclusion of cardio protective also. Further, in vitro and in situ studies were conducted. Better to delete biopharmaceutical from the title, and add ‘preformulation’.

Write in situ uniformly in the manuscript.

Define BCS in abstract also.

Write the solubility and permeability values in the abstract.

Delete ‘we’ from line# 53 and use scientific reporting words.

Define WHO, US-FDA, EMA, etc.

How to define soluble vs solubility as per the regulatory guidelines?

What are the additional steps proved compared with Ref 11 and 12. The compound already proved as neuroprotective? How the authors select the vehicle or suitable dosage for the JM-20 prior to the preformulation studies in ref 11 and 12.

 MDCK, MDCK-MDR1 and Caco-2 – what is the rationale for the selection of this cell lines not justified.

 Calculate the practical log P of the JM-20. Correlate the aqueous solubility vs log P of the compound on permeability and dissolution characteristics.

What is the stability of the JM-20 in the aqueous and organic media?

Write the validation parameters of the JM-20 UV method. Write the macroscopic description of the compound.

Write the significance levels of the MDCK, MDCK-MDR1 and Caco-2 cell lines.

Why metoprolol selected as model or reference compound, why not other neuroprotective compounds or BCS class II drugs.

Author Response

In the abstract and introduction, the authors mentioned anti-ischemic and neuroprotection activity. Rewrite the title with the inclusion of cardio protective also. Further, in vitro and in situ studies were conducted. Better to delete biopharmaceutical from the title, and add ‘preformulation’.

DONE

Write in situ uniformly in the manuscript.

DONE

Define BCS in abstract also.

DONE

Write the solubility and permeability values in the abstract.

DONE

Delete ‘we’ from line# 53 and use scientific reporting words.

DONE

Define WHO, US-FDA, EMA, etc.

Done

World Health Organization (WHO)

U.S. Food and Drug Administration

European Medicines Agency (EMA)

How to define soluble vs solubility as per the regulatory guidelines?

A highly soluble compound as per regulatory guidelines means that the highest dose is soluble (i.e) will be completely dissolved in 250 mL of the prescribed buffers, while the chemical definition of solubility is the achieved concentration at equilibrium in presence of an excess of solid. Highly soluble or high solubility compounds is a regulatory definition that puts the solubility value in the framework of the administered dose in order to know if the dose can be or not dissolved in the intestinal fluid.

What are the additional steps proved compared with Ref 11 and 12. The compound already proved as neuroprotective? How the authors select the vehicle or suitable dosage for the JM-20 prior to the preformulation studies in ref 11 and 12.

In our opinion, this question is answered in previous papers about JM-20. The aim of the study is the biopharmaceutical characterization of a new neuroprotective compound (JM-20). This activity has been amply demonstrated by the developers of the product. Not only references 11 and 12, but also other preclinical studies have been developed. Some of these studies are listed below:

That information and cites have been included in the manuscript

JM-20 protects against 6-hydroxydopamine-induced neurotoxicity in models of Parkinson’s disease: Mitochondrial protection and antioxidant properties

LA Fonseca-Fonseca, VDA da Silva, M Wong-Guerra… - Neurotoxicology, 2021

JM-20 treatment prevents neuronal damage and memory impairment induced by aluminum chloride in rats

M Wong-Guerra, Y Montano-Peguero… - Neurotoxicology, 2021

JM-20, a novel hybrid molecule, protects against rotenone-induced neurotoxicity in experimental model of Parkinson’s disease

LA Fonseca-Fonseca, M Wong-Guerra… - Neuroscience letters, 2019

JM-20 protects memory acquisition and consolidation on scopolamine model of cognitive impairment

M Wong-Guerra, J Jiménez-Martin… - Neurological research, 2019

the authors select the vehicle and the suitable dosage for the JM-20 in a previous study:

Yanier Nuñez-Figueredo et al. JM-20, a novel benzodiazepine–dihydropyridine hybrid molecule, protects mitochondria and prevents ischemic insult-mediated neural cell death in vitro. European Journal of Pharmacology 726 (2014) 57–65.

MDCK, MDCK-MDR1 and Caco-2 – what is the rationale for the selection of this cell lines not justified.

The next sentence has been included in the introduction of the manuscript

Those cell lines are well stablished in vitro model for intestinal human permeability, MDCK has demonstrated a good correlation with human fraction absorbed and MDCK-MDR1 abd Caco-2 cell lines present the added value if expressing P-glycoprotein (CITAS [6–9])

Calculate the practical log P of the JM-20. Correlate the aqueous solubility vs log P of the compound on permeability and dissolution characteristics.

The log P calculated for JM-20 is 3.46 (see Table 1) suggesting that it is a very lipophilic compound. This result is related to the experimental solubility values determined for JM-20 at pH 1.2, 4.5, 6.8 and 7.4 (Table 2), where very low solubility values (9.2 - 30.8 μg/ml) were obtained. At the same time, the high log P value agrees with the permeability values obtained with in vitro cell lines, as well as with the in situ perfusion method, where JM-20 proved to be a highly permeable compound. These results allowed classifying JM-20 as a BCS class 2 compound.

What is the stability of the JM-20 in the aqueous and organic media?

Martinez et al. (36), developed a qualitative stability study of JM-20 in aqueous medium at pH values of 1.2, 4.5 and 6.8, during 48 hours at 37 ± 1°C. At 24 hours some degradation appears at pH 4.5 and 6.8, but the retention time of JM-20 is the same, and only a decrease in the corresponding areas is observed, so it can be deduced that it has not yet undergone changes in its chemical structure. 

According to the owner of the product (technical report), the JM-20 is stable in DMSO and absolute ethanol.

That information has been included

Write the validation parameters of the JM-20 UV method. Write the macroscopic description of the compound.

HPLC method information has been included and validation parameters have been included.

The drug concentration was 100 μM in all experiments and the samples were immediately analyzed by HPLC. The retention time was 2.3 minutes at maximum wavelengths (λmax = 260 nm). The conditions used for HPLC analyses were: mobile phase composed of a mixture of 50% H2O (95% H2O + 0.5% trifluoroacetic) and another 50% HPLC-grade methanol. Stationary phase composed by a phenomenex® KJ0-4282 column, with two filters of 2 μm filled with C-18 microparticles of 40 μm size and a stainless-steel column model Nova Pak C-18 of 150 mm long.

Method was validated and demonstrated to be adequate regarding linearility (r2> 0.999), accuracy (relative error <5%), precision or repeatability (SD ≤ 2%), stability (recovery = 98-102%), filter influence (recovery = 98-102%), specificity (interference < 2%). The lower limit of quantitation of JM-20 was adequated to interpolate the values obtained in  samples

Describe the new compund

Reddish, odorless, crystalline powder.

Write the significance levels of the MDCK, MDCK-MDR1 and Caco-2 cell lines.

We are not sure of fully understanding this question, but in the case the reviewer refers to statistical differences between the ab and ba permeabilities en each cell model the next paragraph has been modified in the discussion

In MDCK-MDR1 cells the permeability in the A-B direction is greater than the permeability in the B-A direction as in the other cell type (significant differences p<0.05). If the drug only passed through passive diffusion, the permeability A-B and B-A would have to be similar, that is, if the transports coincide it is because there is only transport by passive diffusion, but being different implies the existence of transporters. On the other hand, if the B-A permeability of the two types of cells is compared, it is observed that in the MDCK-MDR1 type the permeability is higher (significant differences at p<0.05). This is due to the fact that the JM-20 is a substrate of the P-glycoprotein transporter (MDR1), which is expressed in this cell line and therefore it promotes the passage of the drug to the apical compartment. If the values obtained in the A-B sense are compared, the above conclusion is corroborated, with slightly greater permeability in MDCK cells, since the resistance due to the secretion transporter is not present but statistical differences were not detected.

Why metoprolol selected as model or reference compound, why not other neuroprotective compounds or BCS class II drugs.

The correlation between in vitro or in situ permeabilties and human oral fraction absorbed has been developed with compounds belonging to different therapeutic activities, as the permeation process is related to physicochemical properties of the compound not with the therapeutic activity. Metoprolol is used in the regulatory setting as model or threshold compound to classify in High/low permeability class as it has a permeability value that ensure an oral fraction absorbed higher than 85%. To classify JM20 as High permeability compound we have used metoprolol and eventually other High permeability compound, belonging to BCS Class I or II could be used but the recommended model in the FDA and EMA guidances is metoprolol.

Round 2

Reviewer 1 Report

I can accept.

Author Response

Dear reviewer,

Thanks so much to accept our work to publish in Pharmaceutics

Reviewer 2 Report

The manuscript modified as per the edits suggested.

Author Response

Thanks for your acceptance

This manuscript is a resubmission of an earlier submission. The following is a list of the peer review reports and author responses from that submission.

Round 1

Reviewer 1 Report

In this article, the biopharmaceutical characteristics of JM-20 were evaluated by in vitro membrane permeation and solubility measurements.

(1) >Recently, Martínez et al. obtained similar results for the pKa value (5.25) of JM-20 but using potentiometric titration [30].

[30] is not Martínez et al.

(2) The A-to-B and B-to-A data of metoprolol and other standard compounds are required (e.g., propranolol, ketoprofen, and carbamazepine). To exclude experimental artifact, these control studies are mandatory.

(3) The A to B permeability values of JM-20 is unusually high (223 10-6 cm/sec). Such high permeability is usually not achievable due to the resistance from the unstirred water layer. Actually, it is much higher than the maximum value obtained by the same research group (Figure 4). Please discuss this point.

Reviewer 2 Report

You revised to my comments. Partly, I could understand what you answered to my comments.

However, it was too poor to write all of comments. However, I point out the issues to be addressed.

Major

1) In figure 3, as long as shown in the result that papp (B-A) of JM-20 was higher than that of metoprolol, you must show the inhibitory results. You have mentioned the concern regarding P-gp contribution of JM-20. It must be needed to the inhibitory experiments if you suggest it.

2) In page 9, line 324-326. You should read references I recommend under bellow.

Miyake M et al., Prediction of drug intestinal absorption in human using the Ussing chamber system: A comparison of intestinal tissues from animals and humans. Eur J Pharm Sci., 2017, 96: 373-380.

Hubatsch I et al., Determination of drug permeability and prediction of drug absorption in Caco-2 monolayers. Nat Protoc., 2007, 2(9): 2111-2119.

Minor

i) In page 9, line 295, you should add at hyphen in front of glycoprotein, although I pointed out even the previous comments.

ii) In page 9, line 298, it is like same kinds of point mentioned above.